# The Ubiquitin-Conjugating Enzyme E2 O (UBE2O) and Its Therapeutic Potential in Human Leukemias and Solid Tumors

**DOI:** 10.3390/cancers16173064

**Published:** 2024-09-03

**Authors:** Beatrice Maffeo, Daniela Cilloni

**Affiliations:** Department of Clinical and Biological Sciences, University of Turin, 10043 Orbassano, Italy; daniela.cilloni@unito.it

**Keywords:** UBE2O, UPS, ubiquitin, protein degradation, erythropoiesis, leukemia, solid tumors

## Abstract

**Simple Summary:**

This review provides a comprehensive and detailed explanation of the various functions of the ubiquitin-conjugating enzyme E2 O (UBE2O). It examines UBE2O’s role in a wide range of biological processes under both physiological and pathological conditions. The review also explores UBE2O’s involvement in several human cancers, with a particular focus on its role in leukemias and solid cancers. Furthermore, this review delves into the dual role of UBE2O, not only as a potential therapeutic target but also as a crucial factor in the regulation of the development and progressoin of several human malignancies. In summary, this review underscores UBE2O importance in cancer biology and its potential for future therapeutic applications.

**Abstract:**

Protein degradation is a biological phenomenon essential for cellular homeostasis and survival. Selective protein degradation is performed by the ubiquitination system which selectively targets proteins that need to be eliminated and leads them to proteasome degradation. In this narrative review, we focus on the ubiquitin-conjugating enzyme E2 O (UBE2O) and highlight the role of UBE2O in many biological and physiological processes. We further discuss UBE2O’s implications in various human diseases, particularly in leukemias and solid cancers. Ultimately, our review aims to highlight the potential role of UBE2O as a therapeutic target and offers new perspectives for developing targeted treatments for human cancers.

## 1. Introduction

All cell types during their lifetime must be able to respond to metabolic requirements and environmental changes and, at the same time, maintain their homeostasis. Protein degradation is fundamental for cellular homeostasis and is essential to the control of cell signaling [1]. Much of the control of cell signaling occurs via protein ubiquitination and/or protein phosphorylation. While protein phosphorylation is reversible and allows proteins to be in two or more states and to interchange between them, protein ubiquitination provides the irreversible degradation of targeted proteins, even though it is now appreciated that, under particular conditions, protein ubiquitination could be reversible via the activity of de-ubiquitylating enzymes (DUBs) [2,3,4].

Ubiquitination is one of the most significant cellular post-transcriptional mechanisms and is involved in a wide range of key biological processes and cellular functions, such as protein–protein interaction, protein localization, and expression levels [5]. Furthermore, ubiquitination can mediate DNA repair, gene transcription, inflammatory response, protein trafficking, and angiogenesis [3,6].

Ubiquitin was first discovered in 1975 by Gideon and collaborators [7], who isolated and purified from bovine thymus a polypeptide with a high degree of evolutionary conservation, initially named ubiquitous immunopoietic polypeptide (UBIP). Since then, extensive research has been conducted, and the discovery of the ubiquitin pathway has revolutionized our concept of intracellular protein degradation. Ubiquitin is a polypeptide of 8,5 kDa, composed of 76 residues. In humans, ubiquitin is encoded by four genes: UBA52 and UBA80/RPS27A encode for ribosomal subunits and UBB and UBC encode for polyubiquitin precursors, which are then converted into single ubiquitin by de-ubiquitinating enzymes [8,9]. In addition, other pseudogenes have been identified as potential genes encoding for ubiquitin [10]. During the 1980s, ubiquitin was described as a cofactor of a proteolytic system, and Hershko and coworkers produced the initial understanding of the ubiquitin-mediated protein degradation system, today known as the Ubiquitin Proteasome System (UPS) [11,12]. These discoveries led Aaron Ciechanover, Avram Hershko, and Irwin Rose to win the Nobel prize for chemistry in 2004 [“https://www.nobelprize.org/prizes/chemistry/2004/summary/ (accessed on 28 August 2024)”].

The proteolysis of cellular proteins is a highly complex process regulated and carried out by a complex cascade of enzymes [13]. The degradation of ubiquitin-mediated proteins is a two-step process: (i) multiple ubiquitin molecules are covalently attached to the lysine residue of the target protein; (ii) the tagged protein is degraded by the 26S proteasome. In the first step, the ubiquitin is first activated in its C-terminal Gly by the ubiquitin-activating enzyme, E1. Then, an ubiquitin-conjugating enzyme, E2, facilitates the transfer of the activated ubiquitin from E1 to an ubiquitin ligase enzyme, E3, to which the substrate protein is specifically bound [13,14]. In humans, there are 2 genes encoding for E1s, at least 38 genes encoding for E2s, and more than 600 genes encoding for E3s, thus providing different types of ubiquitination and the ubiquitination of selected, specific substrates [6,15,16,17]. The system has a hierarchical structure, in which a single E1 can activate and transfer ubiquitin to several species of E2s, and each E2 can interact and cooperate with several E3s [13].

The aim of this review is to provide a comprehensive analysis of the functions of UBE2O and its crucial role in many cellular and biological processes. Furthermore, this review addresses the implications of the altered expression of UBE2O in various human leukemias and solid cancers and highlights UBE2O potentialities as a target in several therapeutic strategies.

## 2. The Atypical Ubiquitin-Conjugating Enzyme E2 O: UBE2O

One of the biggest in size E2s known currently is the atypical ubiquitin-conjugating enzyme E2 O (UBE2O), with a molecular weight of 141 kDa [18], while most members of the E2 family have a molecular weight spanning from 20 to 25 kDa [19]. UBE2O was first extracted from rabbit reticulocytes and named E2-230K; subsequently, human UBE2O was cloned from liver tissues [20]. In humans, UBE2O is ubiquitously expressed in all organs and almost all types of tissue, but it is preferentially expressed in the brain, heart, liver tissue, and skeletal muscle [18]. Furthermore, UBE2O is highly conserved in animals and plants [6], suggesting that its function is fundamental in many biological processes. Indeed, since its discovery, several studies have shown a plethora of cellular functions involving UBE2O, from cell cycle and proliferation to bone morphogenesis and erythroid differentiation [6,21].

UBE2O possesses three conserved regions (CR1, CR2, and CR3), a coiled-coil (CC) domain, and a UBC domain, which enables UBE2O to interact with multiple E3 ligases. Interestingly, although it has two nuclear localization sequences (NLSs), UBE2O is mainly localized within the cytoplasm [22,23]. Human UBE2O has 25 different cysteine residues, and 2 of them are mainly responsible of its catalytic activity: Cys617 localized in the CR2 domain is necessary for the ubiquitination of target substrates [24]; Cys1040 in the UBC domain is the E2 active site [22].

Despite being first discovered and described as an E2 enzyme, most of the reported UBE2O substrates are catalyzed by UBE2O in an E3-independent manner [20,21]. Berleth and Pickart described an intramolecular thiol relay mechanism of UBE2O, by which it catalyzed substrates’ ubiquitination [25]. The CR1, CR2, and CC domains found in the N-terminal segment of UBE2O can interact with the E2 active site contained in the C-terminal fragment, thus leading to the mono-ubiquitination of the target (Figure 1, right side). This demonstrates that UBE2O works as an E2/E3 hybrid enzyme and displays E3 ligase activity [6,25,26]. For example, UBE2O was shown to ubiquitinate AMP-activated protein kinase (AMPK), facilitating its degradation via the proteasome and thus activating the mechanistic target of the rapamycin (mTOR) signaling pathway, which includes the hypoxia-inducible factor 1-alpha (HIF-1α), suggesting that UBE2O plays a role in the regulation of the response to hypoxic conditions. Furthermore, as a downstream target of the UBE2O/AMPK/mTOR pathway, MYC transcriptionally promotes UBE2O expression, thus constituting a positive feedback loop that promotes cell proliferation and epithelial–mesenchymal transformation (EMT) in many types of human cancers [27]. Similarly, UBE2O mono-ubiquitinates SMAD6 at lysine 174 during bone morphogenesis, thus reducing SMAD6’s inhibitory effect on the BMP/SMAD downstream pathway, which is essential during bone morphogenesis and heart and central nervous development [28]. Moreover, UBE2O was observed to interfere with the circadian rhythm as well. Mechanistically, the Cys residue in the CR2 UBE2O domain interacts with BMAL1, essential for circadian oscillation [29], thus promoting its ubiquitination and degradation. Hence, UBE2O is a critical regulator of the circadian clock, whose dysregulation is associated with many diseases, including cancer, diabetes, and obesity [24].

Furthermore, UBE2O also displays non-enzymatic functions. By interacting with the TRAF domain of the TNF receptor-associated factor 6 (TRAF6), the N-terminal fragment of UBE2O can block its polyubiquitination on lysine 63 and suppress the activation of NF-κB in a UBE2O dose-dependent manner (Figure 1, left side). The removal of the UBC domain in UBE2O does not impact the inhibition of TRAF6 ubiquitination, thus demonstrating the non-enzymatic functions of UBE2O and indicating that UBE2O is a potent TRAF6 regulator [30].

A plethora of specialized features and multifunctional domains within the UBE2O protein demonstrate the important role of UBE2O in many aspects of cells’ lives. Consequently, alteration of UBE2O expression levels might represent an important factor to be evaluated at the onset and progression of human diseases due to UBE2O’s ability to interfere with a broad spectrum of molecular targets and functions.

## 3. Role of UBE2O in the Hematological Field

The constant removal of surplus and damaged peptides, as well as the maintenance of the protein homeostasis, is essential for proper cellular function. Therefore, alterations in the regulation of protein turnover play a role in the pathogenesis of a number of different types of solid cancers and diseases. Dysregulation of this proteolysis-regulating machinery can result in uncontrolled cell proliferation, accumulation of harmful proteins, and genetic instability, ultimately leading to malignancy [31]. Dysregulations of several members of UPS have been identified in hematological malignancies.

Several therapies that have been developed for the treatment of hematological diseases are based on the manipulation of UPS members [32]. The most relevant example is bortezomib, a selective inhibitor of the 26S proteasome which revolutionized the therapy of multiple myeloma (MM) and Mantle Cell Lymphoma (MCL), leading to it becoming a first-line agent [33]. The deubiquitinating enzyme USP1 was reported to be involved in Fanconi Anemia (FA), a rare hematological disorder caused by the genetic loss of key factors of DNA repair [34]. Moreover, it has been shown that the inhibition of the deubiquitinating enzyme USP10 inhibits proliferation in chronic myeloid leukemia (CML) cells, both in imatinib-sensitive and imatinib-resistance cells [35,36]. On the other hand, many therapeutic strategies directly regulate protein degradation, such as thalidomide analogs. The E3 ubiquitin ligase cereblon (CRBN) was identified as the main binding partner of thalidomide. Fang et al. [37] showed that Lenalidomide, the main clinically used thalidomide-derived compound, acts through CRBN and has cytotoxic effects in cells of both myelodysplastic syndrome (MDS) and acute myeloid leukemia (AML). In addition, in recent years, CRBN-based PROTACs (PROteolysis Targeting Chimeras) have gained attention, showing promising results in inducing the degradation of disease-associated proteins [33].

Here, we report the therapeutic opportunities and strategies based on the activation or inactivation of UBE2O enzyme, describing its potential roles in the hematological field.

### 3.1. UBE2O Regulates Proteome Remodeling during Terminal Erythroid Differentiation

Erythropoiesis consists of the maturation and differentiation of multipotent hematopoietic stem cells to unipotent erythroid progenitors [38]. During terminal differentiation, erythrocyte precursors undergo an extensive remodeling of their organelles, including nucleus, mitochondria, and ribosomes, in order to make space for globin, which is fundamental to oxygen transportation and constitutes ~98% of the cytosol [39]. This reorganization deeply involves the ubiquitination and deubiquitination system [40]. One of the main actors is UBE2O, which is specifically induced and active in terminally differentiating reticulocytes, resulting in massive protein degradation [41,42]. Interestingly, despite UBE2O being the second most abundant mRNA in mouse reticulocytes, it is abundant in reticulocytes, while it is low or absent in undifferentiated cells, suggesting that in erythroid cells, UBE2O is reticulocyte-specific [41]. Nguyen et al. [42] gained mechanistic insight into the function of UBE2O during red blood cells’ (RBCs) maturation (Figure 2A). Taking advantage of the UBE2O-null mouse model, they demonstrated that UBE2O deficiency is associated with a defective mechanism of protein ubiquitination, leading to the extensive intracellular accumulation of ribosomal proteins and the development of microcytic hypochromic anemia in mice. Moreover, UBE2O has been shown to selectively ubiquitinate unassembled α-globin molecules that fail to assemble with β-globin in reticulocytes [43]. Thus, UBE2O plays a prominent role in the regulation of RBC maturation, also acting as a quality control factor in reticulocytes, suggesting a putative role of UBE2O in the treatment of hematological malignancies characterized by ineffective erythropoiesis, such as β-thalassemia and myelodysplastic syndromes.

### 3.2. UBE2O Overexpression Inhibits Acute Myeloid Leukemia Progression

Acute myeloid leukemia (AML) is an aggressive hematological malignancy characterized by the accumulation of immature myeloid progenitors [44]. Although very few studies have investigated the involvement of UBE2O in acute leukemias, many studies have addressed the role of the bone marrow (BM) microenvironment in supporting leukemia, demonstrating that the BM microenvironment can be remodeled to support leukemogenesis and impede normal hematopoiesis [45,46]. Tian et al. identified around 1000 genes with an altered expression in AML-mesenchymal stromal cells (MSCs) and selected UBE2O as the target gene of their study [47,48]. The lentiviral overexpression of UBE2O in AML-MSCs demonstrated that high levels of UBE2O inhibit the proliferation of MSCs via the deactivation of the NF-kB pathway. Furthermore, they showed that UBE2O-overexpressing cells, due to a lower adhesion capability, led to the decreased growth of AML cells and the prolonged survival of mice (Figure 2B). These results provide the theoretical basis for a BM microenvironment-based therapeutic strategy to control disease progression. The demonstration that UBE2O overexpression in MSCs inhibits the proliferation of leukemic cells suggests that the induction of UBE2O overexpression could be investigated as a strategy to control disease progression.

### 3.3. UBE2O Knockdown Reduces Cell Proliferation in KMT2A Rearranged Leukemias

Contrary to the hematological malignancies evaluated so far, for which the upregulation of UBE2O could represent a novel therapeutic strategy, other types of leukemia would benefit from the inhibition of UBE2O.

KMT2A (also known as MLL, mixed-lineage leukemia, or myeloid–lymphoid leukemia) belongs to the group of the KMT genes and encodes a DNA-binding protein methylating histone H3 lys4 (H3K4), the lysine methyltransferase 2A [49,50]. Rearrangements involving KMT2A have been shown to occur in precursors in B-acute lymphoblastic leukemia (B-ALL), T-acute lymphoblastic leukemia (T-ALL), acute myeloid leukemia (AML), myelodysplastic syndrome (MDS), mixed-lineage (biphenotypic) leukemia (MPAL), and secondary leukemia [51]. The KMT2A rearrangements are typically associated with poor prognosis and a disease progression often accompanied by a drift or switch in lineage, such as from lymphoid to myeloid [52]. The rearrangements of KMT2A, located on chromosome 11q23s, usually occur as a single mutation [53,54]; however, they can also be present with a cooperative mutation, especially PI3K-RAS, KRAS, NRAS, and TP53 [51,53,55,56].

Taking advantage of multidimensional protein identification technology (MudPIT) assays and coimmunoprecipitation assays, Liang et al. [23] showed that UBE2O is the most abundant protein specifically interacting with wild-type KMT2A. The wilt-type KMT2A is less stable than the chimeric KMT2A, thus suggesting that stabilizing the KMT2A protein in leukemic cells could have a therapeutic potential [23,57]. Specifically, they demonstrated that interleukin-1 β (IL-1β) increases the KMT2A-UBE2O interaction and induces the polyubiquitination of wild-type KMT2A, leading to its degradation. Furthermore, the interleukin-1 receptor-associated kinases (IRAKs) could phosphorylate UBE2O, thereby enhancing its interaction with wild-type KMT2A (Figure 2D). Hence, the downregulation of UBE2O levels could potentially improve the stability of wild-type KMT2A. RNA-seq analysis of UBE2O knocked-down cells revealed the downregulation of a plethora of genes, also associated with cell cycle activation or cellular response to growth factor stimulus [6]. In conclusion, in KMT2A-related diseases, UBE2O depletion increases the stability of wild-type KMT2A and consequently downregulates a common subset of KMT2A chimera target genes and reduces leukemic cells’ proliferation

### 3.4. UBE2O Induces Apoptosis in Multiple Myeloma Cells

Multiple myeloma (MM) is one of the most commonly diagnosed blood cancers [58]. One of the genes commonly involved in the pathophysiology of MM is the transcription factor c-Maf that belongs to the Maf family. c-Maf is overexpressed in more than 50% of MM patients and in MM cell lines [59,60]. Zhang et al. [61] observed that c-Maf significantly enhances myeloma cell proliferation in vitro and tumor formation in vivo. UBE2O interacts with c-Maf and mediates the polyubiquitination of c-Maf at K48 inducing its degradation through the ubiquitin proteasome pathways [61,62]. Moreover, UBE2O also downregulates the transcriptional activity of c-Maf and the expression of c-Maf downstream genes, including cyclin D2, integrin β7, CCR1 (C-C Motif Chemokine Receptor 1), and ARK5 (AMPK-related protein kinase 5), which are responsible for cell cycle progress, MM cell proliferation, and myeloma cell invasion [62,63]. c-Maf destabilization and degradation impair the growth of MM. Furthermore, UBE2O has been found to be downregulated in MM cells, and the restoration of UBE2O levels and activity induces MM cell apoptosis and suppresses cell proliferation both in vitro and in vivo, suggesting that UBE2O acts as a tumor suppressor against MM and might therefore be a potential therapeutic strategy for the treatment of multiple myeloma (Figure 2C) [62].

## 4. UBE2O Acts as a Pro-Oncogenic Factor in Many Human Cancers

The deregulation of UBE2O and its subsequent abnormal expression occur in many types of tumor and are associated with several human diseases [21,27,64,65] (Figure 3). UBE2O is localized in the 17q25 chromosome region, which has been found to be amplified in different cancers, including breast, prostate, gastric, kidney, and ovarian cancers [21,66,67,68]. All this evidence supports the strong connection that exists between UBE2O deregulation and tumor initiation and progression.

In recent years, much attention has been paid to evaluating the role of UBE2O overexpression in the pathogenesis and progression of breast and prostate cancers. In 2017, Vila et al. [66] demonstrated that high levels of UBE2O promote tumor initiation in mouse models of breast and prostate cancers in an AMPKα2-dependent manner. Mechanistically, UBE2O ubiquitinates AMPKα2, facilitating its degradation, which in turn determines the activation of the mTORC1 signaling pathway. The UBE2O/AMPKα2/mTORC1 axis favors the transcriptional activity of the oncoprotein MYC, generating a positive feedback loop that promotes cancer cell proliferation and epithelial–mesenchymal transformation (EMT) [27] and endows BC cells with cancer stemness properties. Furthermore, the pharmacological inhibition of UBE2O reduces its protumor activity through the restoration of AMPKα2 [27,66].

Furthermore, the in silico analysis of the TCGA database demonstrated that UBE2O amplification is frequently found in many types of gastric and lung cancers [6]. In lung cancer, which is the deadliest malignancy worldwide, UBE2O has been shown to play a putative essential role in radioresistance. Huang et al. [69] showed that UBE2O ubiquitinates MAX interactor 1 (Mxi1), whose degradation results in reduced radiosensitivity to squamous cell carcinoma and malignant lymphoma in mouse models [70]. Consistently, UBE2O overexpression promotes lung cancer proliferation and radioresistance and predicts poor-prognosis patients through the negative regulation of Mxi1 [69]. Hence, the genetical or pharmacological inhibition of UBE2O could be a potential strategy for treating lung cancer.

Recent studies have also highlighted the potential of targeting the UBE2O/HADHA axis in the treatment of hepatocellular carcinoma (HCC), the most common primary malignant liver tumor and the sixth highest cause of cancer-related death [71]. HADHA is a mitochondrial β-oxidation enzyme and is downregulated in HCC. UBE2O expression negatively correlates with HADHA levels, as UBE2O can mediate the ubiquitination and degradation of HADHA [72]. Through the regulation of HADHA, UBE2O modulates lipid metabolic reprogramming and promotes HCC with poor survival. The liver-specific deletion of UBE2O inhibits HCC growth and metastasis and is sufficient to prevent metabolic reprogramming in mice models [72]. In addition, novel integrating data from proteomic, mass spectrometry, and survival analysis identified the interferon-induced protein (IFIT3), a mediator of interferon (IFN) signaling and inhibitor of cell proliferation and migration [73], as a ubiquitination substrate of UBE2O that recognizes the lysine 236 of IFIT3. In HCC, high expression of IFIT3 increases the effectiveness of IFN therapy by upregulating the IFN-α signaling pathway and response [74]. Li et al. [75] demonstrated that the knockdown of UBE2O enhances the efficacy of IFN-α signaling, suggesting that targeting UBE2O could be a promising strategy to increase the therapeutic effect of interferon-α.

These observations highlighted the pro-oncogenic role of UBE2O and support the concept of UBE2O targeting as a potential therapeutic strategy in diseases where metabolic reprogramming and proteome remodeling play a role.

## 5. UBE2O’s Role in Non-Oncologic Diseases

In addition to hematological diseases and cancers, UBE2O has been found to be involved in many other malignancies (Table 1). In 2019, Vila et al. [76] demonstrated that UBE2O’s expression is universally upregulated in animal models exhibiting insulin resistance and metabolic disorders and significantly elevated in obese subjects with type 2 diabetes. AMPK is a key sensor and a crucial regulator of metabolism and energy balance [77]. As already reported, UBE2O ubiquitinates AMPKα2 for its degradation; therefore, UBE2O inhibition could improve metabolic balance through AMPK pathway activity. Through UBE2O knockout models, they managed to improve insulin sensitivity in diet-induced type 2 diabetes mice and demonstrated that UBE2O ablation protects mice against diet-induced obesity and metabolic syndrome. This evidence suggests that the development of drugs interfering with UBE2O-AMPK interaction or the pharmacological inhibition of UBE2O activity could represent a powerful strategy for contributing to muscle insulin sensitivity, ameliorating diabetes and metabolic health.

Also, dysfunctions in the UPS are associated with the accumulation of misfolded or damaged proteins in the brain, which is the hallmark of age-related neurodegenerative diseases, such as Alzheimer disease (AD) and Parkinson’s and Huntington’s diseases. The peak expression of UBE2O happens in the second week postanal, and then its expression is gradually reduced with age, consistent with the decline in UPS with aging. Cheng et al. [78] recently demonstrated that UBE2O is reduced in the cortex and hippocampus of AD mice and reduced levels of UBE2O are associated with increased neuronal death. Mechanistically, UBE2O should contribute to reduced amyloid-β (Aβ) deposit, but the accumulation of the Aβ protein precursor (AβPP) with a pathogenic mutation for AD reduces the expression of UBE2O. These data suggest that UBE2O is dysregulated in AD neurons, and an age-associated reduction in UBE2O may facilitate neuronal death in AD and indicate that increasing UBE2O expression in neurons may have a therapeutic potential for AD.

## 6. Conclusions and Future Prospects

The primary role of protein degradation in cell homeostasis and disease onset has become much more evident in recent years, attracting the interest of many studies. The biological and clinical relevance of E2s to the pathogenesis and progression of diseases and cancer suggest that E2s potentially hold great therapeutic promises as druggable targets [21,79]. Despite recent progress in the development of additional small-molecule E2 inhibitors [80,81], no such E2-targeting therapy has yet made its way to clinical trials [21]. Arsenic, which can crosslink adjacent cysteines within the catalytic approach of UBE2O, could serve as the basis of an alternative approach to inhibiting E2 activity and is currently being tested against various forms of cancer in clinical trials (https://clinicaltrials.gov) [21,66]. It is also expected that E2-targeting therapeutics will be more efficacious against diseases and cancer when used in combination with current chemotherapy regimens.

While, on the one hand, the development of drugs designed to block UBE2O activity could provide a relevant mechanism for the inhibition of the progression of many tumors and for an improvement in insulin sensitivity and systemic physiology, on the other hand, UBE2O is required for erythropoiesis and has showed a favorable role in some hematological diseases.

Although there is currently no data supporting the use of UBE2O as a prognostic marker or for predicting therapy response in solid tumors or hematological malignancies, the role of this enzyme suggests that it could potentially serve as a biomarker for therapy response in the future.

In this regard, the ongoing study in our laboratory identifies UBE2O as a potential biomarker of ineffective erythropoiesis in patients with MDS. Additionally, our data suggest that luspatercept, a drug approved for the treatment of anemia in MDS, acts through the upregulation of UBE2O, leading to a stimulation of effective erythropoiesis. If these findings are confirmed, UBE2O could serve as a biomarker for selecting patients who would benefit from treatment with luspatercept [82].

Finally, the final goal of these studies is to target UBE2O through specific treatments. To date, no selective inhibitors of UBE2O have been developed, but the potential to create targeted drugs makes this an increasingly attractive target. There are ongoing studies in solid tumors testing molecules that act as selective inhibitors of ubiquitin-activating enzymes. One such inhibitor is TAK-243 which targets UAE (UBA1) and blocks ubiquitin conjugation, thereby disrupting both monoubiquitin signaling and global protein ubiquitination. This drug is currently undergoing clinical trials for advanced cancers (ClinicalTrials.gov, NCT06223542).

Finally, there is the possibility of targeting ubiquitin-specific protease 1 (USP1), which is an enzyme capable of deubiquitination. Several trials are currently underway with TNG348, a USP1 inhibitor, both alone and in combination with Olaparib in patients with BRCA1/2 mutant solid tumors (ClinicalTrials.gov, NCT06065059).

Although further studies are needed, UBE2O regulation could be a promising strategy in the treatment of different human neoplasia.

## Figures and Tables

**Figure 1 cancers-16-03064-f001:**
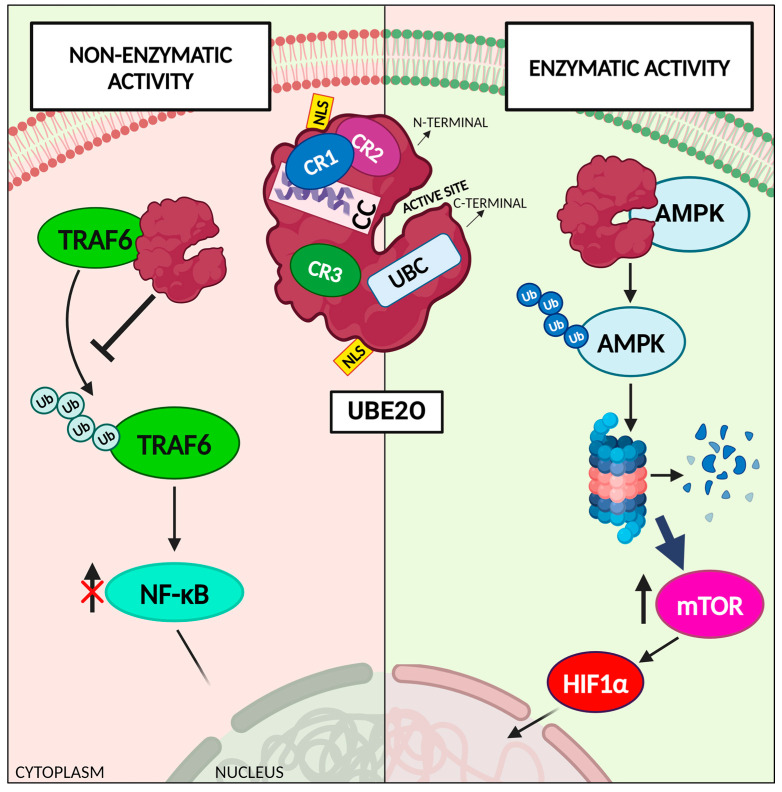
UBE2O exerts both enzymatic functions and non-enzymatic functions. The most studied and reported function of UBE2O is that as a hybrid E2/E3 enzyme, it catalyzes substrates ubiquitination. Nonetheless, UBE2O has been shown to also exert non-enzymatic functions. Figure 1 shows two examples of different functions of UBE2O. The enzymatic function is explained through the depiction of the AMPK-mTOR pathway, in which UBE2O catalyzes the ubiquitination of AMPK, thus inducing its degradation and the activation of mTOR and subsequently of HIF1α. The non-enzymatic function of UBE2O is represented through the schematization of the interaction of the UBE2O N-terminal fraction with TRAF6, thus impeding its ubiquitination and negatively regulating the NF-κB pathway. AMPK: 5′ adenosine monophosphate-activated protein kinase; CC: coiled-coil region; CR1/2/3: conserved region 1/2/3; HIF1α: hypoxia-inducible factor 1-alpha; mTOR: mammalian target of rapamycin; NLS: nuclear localization sequence; NF-kB: nuclear factor kappa-light enhancer of activated B cells; TRAF6: tumor necrosis factor receptor-associated factor 6; UB: ubiquitin; UBE2O: ubiquitin-conjugating enzyme E2 O; UBC: ubiquitin-conjugating core domain. Created with “www.BioRender.com (accessed on 28 August 2024)”.

**Figure 2 cancers-16-03064-f002:**
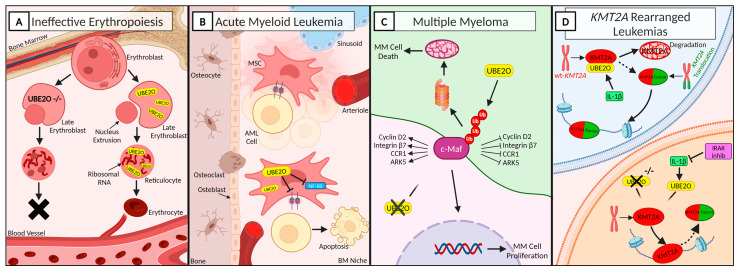
UBE2O in hematology under physiological and pathological conditions. A fine regulation of protein turnover and degradation is essential for maintaining cells’ homeostasis. Regarding an alteration at this level, UBE2O is important in many aspects of hematology. (**A**) UBE2O is essential for proper erythroid differentiation. Hence, UBE2O silencing leads to the incapability of generating red blood cells. UBE2O: ubiquitin-conjugating enzyme E2 O. (**B**) Overexpression of UBE2O in AML cells inhibits leukemic cells’ proliferation and induces apoptosis. AML: acute myeloid leukemia; BM: bone marrow; MSCs: mesenchymal stromal cells; NF-κB: nuclear factor kappa B; (**C**) UBE2O activity promotes c-Maf polyubiquitination and the subsequent apoptosis of MM cells. ARK: AMPK-related protein kinase 5; CCR1: C-C Motif Chemokine Receptor 1; MM: multiple myeloma. (**D**) UBE2O depletion promotes the stability of wild-type KMT2A in hematological neoplasia characterized by KMT2A rearrangements. IL-1β: interleukin-1β; IRAKs: interleukin-1 receptor-associated kinases; KMT2A: lysine methyltransferase 2A; WT: wild-type. Created with “www.BioRender.com (accessed on 28 August 2024)”.

**Figure 3 cancers-16-03064-f003:**
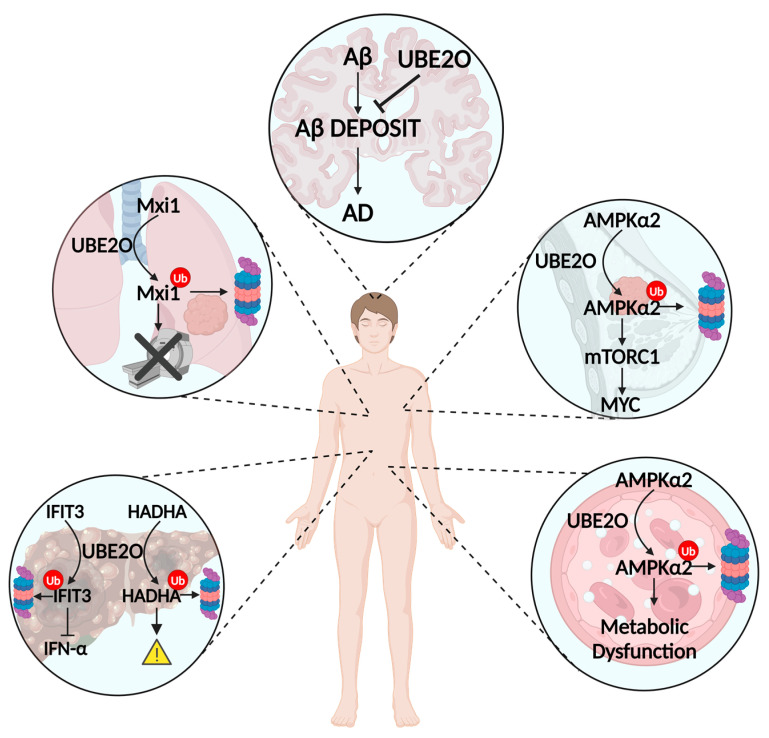
UBE2O in non-blood diseases. Overview of UBE2O mechanism of action in non-hematological diseases: lower left: hepatocellular carcinoma; upper left: lung cancer; center: Alzheimer disease; upper right: breast cancer; lower right: type 2 diabetes. Aβ: amyloid-β; AD: Alzheimer’s disease; AMPKα2: AMP-activated protein kinase alpha 2; HADHA: hydroxyacyl-CoA dehydrogenase; IFIT3: interferon-induced protein; IFN-α: interferon-alpha; mTORC1: mammalian target of rapamycin complex 1; Mxi1: MAX interactor 1; MYC: also known as cMyc. Created with “www.BioRender.com (accessed on 28 August 2024)”.

**Table 1 cancers-16-03064-t001:** The dual role of UBE2O in different human diseases. Schematic classification of UBE2O downregulation and/or upregulation beneficial effects in different human leukemias, solid tumors, and other diseases. Aβ: amyloid-β; AML: acute myeloid leukemia; AMPKα2: AMP-activated protein kinase alpha 2; HADHA: hydroxyacyl-CoA dehydrogenase; IFN-α: interferon-alpha; mTORC1: mammalian target of rapamycin complex 1; Mxi1: MAX interactor 1; MM: multiple myeloma; NF-κB: nuclear factor kappa B; KMT2A: lysine methyltransferase 2A; WT: wild-type.

	UBE2O Regulation	Effects
Leukemic diseases		
Myelodysplastic syndromes and β-thalassemia	Upregulation	α-globin ubiquitination; promotion of proteome remodeling during RBC maturation
Acute myeloid leukemia	Upregulation	NF-κB pathway inhibition; reduced AML cells’ proliferation
Multiple myeloma	Upregulation	c-Maf ubiquitination; increased MM cell death
*KMT2A* rearranged leukemias	Downregulation	Increased WT-KMT2A stability; reduced leukemic cells’ proliferation
Solid tumors		
Breast and prostate cancer	Downregulation	Reduced AMPKα2 ubiquitination; reduced activation of mTORC1 pathway; reduced cancer cell proliferation
Lung cancer	Downregulation	Decreased Mxi1 ubiquitination; reduced radioresistance; reduced cancer proliferation
Hepatocellular carcinoma	Downregulation	Reduced HADHA ubiquitination; increased IFN-α signaling efficacy; reduced cancer cells survival.
Non-oncologic diseases		
Metabolic disorders	Downregulation	Reduced AMPKα2 ubiquitination; increased insulin sensitivity
Alzheimer diseases	Overexpression	Reduced Aβ deposit; reduced neuronal cells death

## Data Availability

No new data were created or analyzed in this study. Data sharing is not applicable to this article.

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
