# Peer review of "The Ubiquitin-Conjugating Enzyme E2 O (UBE2O) and Its Therapeutic Potential in Human Leukemias and Solid Tumors"

_cancers, 2024, doi:10.3390/cancers16173064_

Round 1

Reviewer 1 Report

Comments and Suggestions for Authors

The manuscript titled “The Controversial Role of Ubiquitin-Conjugating Enzyme E2 O (UBE2O) in Human Leukemias and Solid Tumors” provides a comprehensive review of the role of the ubiquitin conjugating enzyme UBE2O within the UPS system in the context of biology and disease. The content offers valuable insights into the critical role of this unique E2-E3 hybrid enzyme in maintaining protein homeostasis under physiological and disease conditions. The review aligns well with the theme of the special issue “how ubiquitin and ubiquitination affect cancer progression”, making it a valuable contribution to the field. However, some sections lack clarity, with instances where the writing did not convey the original research findings, leading to potential misunderstandings. The review needs revision to address these issues.

·      Page 1 Title: a typo in the title. It should be “Ubiquitin-Conjugting Enzyme UbE2 O”, not “Conjugtin Enzyme”

·      Page 2 line 52-54: “The activated ubiquitin is then transferred from E1 to a Cys residue of an ubiquitin-conjugating enzyme, E2, which finally transfers ubiquitin from E1 to an ubiquitin-ligase enzyme, E3, to which the substrate protein is specifically bound.” Consider to reword the statement about the process of protein ubiquitination, which is not clearly or accurately presented. As an example, most RING type E3 ligases do not directly receive Ub from E2. They bring together an E2 and the substrate lysine to facilitate the transfer of Ub from the E2 to the substrate lysine.

·      Page 3 Figure 1: the role of UBE2O in the regulation of AMPK-mTOR-HIF1a axis.   Consider to revise the figure or the text as the role of HIF1a is not mentioned in the text.

·      Page 7 line 276-279: Revise the text indicated to align with the research findings. “Huang et al [68] showed that UBE2O ubiquitinates MAX interactor 1 (Mxi1), whose degradation results in increased sensitivity to squamous cell carcinoma and malignant lymphoma in mouse models”. It is clear from the reference cited that E2O controls the accumulation of Mxi1. Overexpression of Mxi1 enhances radiosensitivity while degradation of Mxi1 promotes cancer progression and radioresistence.

o   Huang, Y.; Yang, X.; Lu, Y.; Zhao, Y.; Meng, R.; Zhang, S.; Dong, X.; Xu, S.; Wu, G. UBE2O Targets Mxi1 for Ubiquitination and Degradation to Promote Lung Cancer Progression and Radioresistance. Cell Death Differ 2021, 28, 671–684,

Comments on the Quality of English Language

·      Proofreading: Consider to conduct a thorough proofreading to correct typos and grammar errors.

o   Page 1, line 29 should be “gene expression”

o   Page 2, line 48 should be “the degradation of proteins mediated by ubiquitin”

o   Page 2, line 49 should be “a two-step process”

o   Page 4, line 123 should be “demonstrate”…. “ cell life”

o   Page 4, line 153 should be “field”

o   Page 6, line 225 – 236 should be “wild-type”

o   Page 7 line 293 should be “lysine 236”

Author Response

Dear Editor and Reviewers,

We sincerely appreciate the effort that have been invested in providing constructive feedback on our manuscript entitled “The Controversial Role of Ubiquitin-Conjugating Enzyme E2 O (UBE2O) in Human Leukemias and Solid Tumors”. We have carefully considered each comment and suggestion and have made related revisions to improve the clarity, accuracy, and overall quality of the work. In this rebuttal we address the points raised by the reviewers, indicating the specific changes made to the manuscript.

Below, we provide a point-by-point response to each of the reviewers’ comments.

Answer to Reviewer 1

  1. Page 1 Title: a typo in the title. It should be “Ubiquitin-Conjugating Enzyme UbE2 O”, not “Conjugtin Enzyme”

The typo has been corrected

  1. Page 2 line 52-54: “The activated ubiquitin is then transferred from E1 to a Cys residue of an ubiquitin-conjugating enzyme, E2, which finally transfers ubiquitin from E1 to an ubiquitin-ligase enzyme, E3, to which the substrate protein is specifically bound.”Consider to reword the statement about the process of protein ubiquitination, which is not clearly or accurately presented. As an example, most RING type E3 ligases do not directly receive Ub from E2. They bring together an E2 and the substrate lysine to facilitate the transfer of Ub from the E2 to the substrate lysine.

The sentence in page 3 line 52-54 has been rewritten as follow: “Then, an ubiquitin-conjugating enzyme, E2, facilitates the transfer of the activated ubiquitin from E1 to an ubiquitin-ligase enzyme, E3, to which the substrate protein is specifically bound.”

  1. Page 3 Figure 1: the role of UBE2O in the regulation of AMPK-mTOR-HIF1a axis.  Consider to revise the figure or the text as the role of HIF1a is not mentioned in the text.

We revised the text page 4 line 116-120 as follow: “UBE2O was shown to ubiquitinate the AMP-activated protein kinase (AMPK), facilitating its degradation via the proteasome and thus activating the mechanistic target of rapamycin (mTOR) signaling pathway, which includes the hypoxia-inducible factor 1-alpha (HIF-1α), suggesting a role for UBE2O in the regulation of the response to hypoxic conditions.”

  1. Page 7 line 276-279: Revise the text indicated to align with the research findings. “Huang et al [68] showed that UBE2O ubiquitinates MAX interactor 1 (Mxi1), whose degradation results in increased sensitivity to squamous cell carcinoma and malignant lymphoma in mouse models”. It is clear from the reference cited that E2O controls the accumulation of Mxi1. Overexpression of Mxi1 enhances radiosensitivity while degradation of Mxi1 promotes cancer progression and radioresistence.

o   Huang, Y.; Yang, X.; Lu, Y.; Zhao, Y.; Meng, R.; Zhang, S.; Dong, X.; Xu, S.; Wu, G. UBE2O Targets Mxi1 for Ubiquitination and Degradation to Promote Lung Cancer Progression and Radioresistance. Cell Death Differ 2021, 28, 671–684,

The text has been revised and aligned with the reference.

Reviewer 2 Report

Comments and Suggestions for Authors

This review summarizes recent studies on the biological role of UBE2O in cancer progression. Overall, the manuscript is well-written, but there are several minor grammatical issues.

line 156: It should read “Figure 2.”

line 241: It should be “overexpressed.”

line 268: The word “then” may need to be deleted.

Line 293: It should be “lysine.”

Line 323: It should say “reduces the expression…”"

Author Response

  1. line 156: It should read “Figure 2.”
  2. line 241: It should be “overexpressed.”
  3. line 268: The word “then” may need to be deleted.
  4. Line 293: It should be “lysine.”

All the suggested corrections have been reviewed and the text modified, as indicated.

Reviewer 3 Report

Comments and Suggestions for Authors

The manuscript ID cancers-3122590 compiles relevant information as a mini-review about the role of ubiquitin-conjugatin enzyme E2 O (UBE2O) in human cancer and solid tumors. The manuscript is very interesting and incorporates relevant information for the topic and the readership. Some points should be addressed before further consideration.

1. Title: Revise the title, specifically in the term "controversial," since this term can confuse the readers. I could understand the plausible authors' aim with such a term, but I think their review is better oriented to define and clarify the role of UBE2O. Indeed, any section develops an idea related to a controversial role. Therefore, I recommend that the authors find another term to better describe their aim without a potentially confusing background.

2. Line 10: specify that the review type is related to a narrative one.

3. Line 14: Add a conclusive sentence in the abstract to state the scope of this review.

4. Line 59: I consider that a paragraph defining the aim and scope should be added to section 1 as an introduction requirement. This can provide a good visualization of the rest of the information compiled in this review.

5. A Table summarizing the role of UBE2O in different conditions, including particular specifications and observations, can be added to this review for easy interpretation for readers.

6. A more detailed section comprising outlook/perspectives/future studies related to this topic can be added to state potential avenues for further research.

Comments on the Quality of English Language

The manuscript can be deeply scrutinized to revise some grammar and stylistic issues.

Author Response

  1. Title: Revise the title, specifically in the term "controversial," since this term can confuse the readers. I could understand the plausible authors' aim with such a term, but I think their review is better oriented to define and clarify the role of UBE2O. Indeed, any section develops an idea related to a controversial role. Therefore, I recommend that the authors find another term to better describe their aim without a potentially confusing background.

The titled has been modified as indicated and rewritten as follow: “The Ubiquitin-Conjugating Enzyme E2 O (UBE2O) and its Therapeutic Potential in Human Leukemias and Solid Tumors”.

  1. Line 10: specify that the review type is related to a narrative one.

In the abstract, line 16, we specified that the review type.

  1. Line 14: Add a conclusive sentence in the abstract to state the scope of this review.

A conclusive sentence with the scope of the review has been added to the abstract.

  1. Line 59: I consider that a paragraph defining the aim and scope should be added to section 1 as an introduction requirement. This can provide a good visualization of the rest of the information compiled in this review.

A section with the aim and scope of the review has been added at the end of chapter 1, as indicated.

  1. A Table summarizing the role of UBE2O in different conditions, including particular specifications and observations, can be added to this review for easy interpretation for readers.

An additional table that summarizes the role of UBE2O under different conditions has been added to the text. Table 1, page 11-12.

  1. A more detailed section comprising outlook/perspectives/future studies related to this topic can be added to state potential avenues for further research.

A section with future perspectives of this research has been added to chapter 6: “Conclusion and future prospects”.

Reviewer 4 Report

Comments and Suggestions for Authors

Beatrice Maffeo and Daniela Cilloni present a high-quality and well-written review manuscript focused on the controversial role of Ubiquitin-Conjugating Enzyme E2 O (UBE2O) in human leukemias and solid tumors.

The authors investigate the multifaceted roles of UBE2O in cellular processes and its implications in various types of cancer, particularly leukemias and solid tumors. They provide an in-depth overview of the ubiquitination system, followed by a detailed analysis of UBE2O’s enzymatic and non-enzymatic functions. The review comprehensively covers UBE2O’s involvement in the regulation of erythropoiesis, the progression of acute myeloid leukemia (AML), and its potential oncogenic role in various solid tumors.

Authors investigated both enzymatic and non-enzymatic functions of UBE2O. Then they covered UBE2O involvement in erythropoiesis and hematological malignancies (AML). They continued with UBE2O oncogenic role in solid tumors - in particular, breast and prostate cancers, lung cancer and radioresistance.

Finally, the authors conclude with a discussion on the dual nature of UBE2O as both a potential therapeutic target and a critical regulator in several diseases. The authors emphasize the need for further research to elucidate the precise mechanisms by which UBE2O contributes to tumorigenesis and to explore its potential as a target for cancer therapy. The authors proposed that further research should focus on developing inhibitors that target UBE2O’s oncogenic functions, particularly in solid tumors. Conversely, enhancing UBE2O activity might be beneficial in hematological malignancies like AML, where it acts as a tumor suppressor.

Overall, the manuscript presents a comprehensive review on the role of Ubiquitin-Conjugating Enzyme E2 O (UBE2O) in various cancers, providing critical insights into its dual functionality and potential therapeutic implications. The manuscript is highly valuable for the scientific community and should be accepted for publication.

======================

Other comments to authors:

1) Please check for typos throughout the manuscript.

Minor typos to consider:

- The word "survivor" should be corrected to "survival" in the sentence: "Protein degradation is a biological phenomena essential for cellular homeostasis and survivor"​.

- The word "process" should be "processes" in the sentence: "...highlight the role of UBE2O in many biological and physiological process"​.

- The sentence "UBE2O regulation could be a promising strategy in the treatment of different human neoplasia" should have "neoplasia" corrected to "neoplasias"​.

- In the caption for Figure 1, the phrase "keeping cells’ homeostasis" should be corrected to "maintaining cellular homeostasis"​.

2) Please improve figures/tables where appropriate.

3) Authors are encouraged to slightly expand the discussion on the clinical implications of UBE2O, particularly its potential as a biomarker for stratifying patients in personalized medicine. Additionally, further elaboration on how UBE2O could be targeted therapeutically in both hematological malignancies and solid tumors would strengthen the manuscript.

4) Lines 139-142. With regards to targeting USP deubiquitinating enzyme - authors are kindly encouraged to cite the following article that describes novel approached in developing USP inhibitors in cancer immunotherapy. DOI: 10.3390/cancers14225539

Author Response

  1. Please check for typos throughout the manuscript.

Minor typos to consider:

- The word "survivor" should be corrected to "survival" in the sentence: "Protein degradation is a biological phenomena essential for cellular homeostasis and survivor"​.

- The word "process" should be "processes" in the sentence: "...highlight the role of UBE2O in many biological and physiological process"​.

- The sentence "UBE2O regulation could be a promising strategy in the treatment of different human neoplasia" should have "neoplasia" corrected to "neoplasias"​.

- In the caption for Figure 1, the phrase "keeping cells’ homeostasis" should be corrected to "maintaining cellular homeostasis"​.

All the minor typos have been corrected.

  1. Please improve figures/tables where appropriate.

An additional table that summarizes the role of UBE2O under different conditions has been added to the manuscript.

  1. Authors are encouraged to slightly expand the discussion on the clinical implications of UBE2O, particularly its potential as a biomarker for stratifying patients in personalized medicine. Additionally, further elaboration on how UBE2O could be targeted therapeutically in both hematological malignancies and solid tumors would strengthen the manuscript.

A short section with future perspective and the clinical potentialities of UBE2O has been added to the chapter 6 “Conclusion and Future Prospects”.

  1. Lines 139-142. With regards to targeting USP deubiquitinating enzyme - authors are kindly encouraged to cite the following article that describes novel approached in developing USP inhibitors in cancer immunotherapy. DOI: 10.3390/cancers14225539

The reference suggested has been added to the revised manuscript.